Integrative analysis of microRNA and mRNA expression profiles in fetal rat model with anorectal malformation

Xiao Hui 1 2
Huang Rui huangrui_alice@126.com 3
Diao Mei 1
Li Long 1
Cui XiaoDai 3
1 Department of Pediatric Surgery, Capital institute of Pediatric , Beijing , China
2 Graduate School of Peking Union Medical College, Chinese Academy of Sciences , Beijing , China
3 Department of Key Laboratory, Capital Institute of Paediatric , Beijing , China
Johnson Charles
Electronic publication date: 2018 Oct 24
Publication date: 2018
Volume: 6
Electronic Location ID: e5774
Received 2018 Jun 4; Accepted 2018 Sep 14
Copyright: ©2018 Xiao et al.
Copyright year: 2018
Copyright holder: Xiao et al.
License: This is an open access article distributed under the terms of the Creative Commons Attribution License, which permits unrestricted use, distribution, reproduction and adaptation in any medium and for any purpose provided that it is properly attributed. For attribution, the original author(s), title, publication source (PeerJ) and either DOI or URL of the article must be cited.
License URL: https://creativecommons.org/licenses/by/4.0/

Keywords: Anorectal malformation, microRNA, mRNA, Fetal rats

Funding: The authors received no funding for this work.

==============================
Background

Anorectal malformations (ARMs) are the most common congenital malformation of the gut, and regulated by multiple signal transduction pathways. The microRNA (miRNA) expression profiles and their biologial functions in anorectal malformations (ARMs) remain unclear. The aim of our study was to evaluate miRNA and mRNA expression profiles in the ARM rats.

Methods and Materials

ARM was induced with ethylenethiourea (ETU) on gestational day 10. Cesarean deliveries were performed to harvest the embryos on gestional day 20. For the extraction of total RNA, 1 cm terminal hindgut samples were obtained from three fetal rats that had similar weights. The microarrays and quantitative RT-PCR analysis were conducted to evaluate the miRNA and mRNA expression profiles in normal fetal rats (n = 3) and ARM fetal rats (n = 3).

Results

In total, 33 miRNAs and 772 mRNAs were significantly and differentially expressed in terminal hindgut tissues of ARM group versus control group. Functional annotation was performed to understand the functions and pathways of differentially expressed mRNAs. Also, we constructed a miRNA-target gene regulatory network including 25 differentially expressed miRNAs and 76 mRNAs. Furthermore, the credibility of the microarray-based results were validated by using qRT-PCR.

Conclusion

The miRNA and mRNA expression in terminal hindgut tissue of ARM fetal rats might provide a basis for further research on the pathogenesis of ARMs.

Introduction

Anorectal malformations (ARMs) are among the most common congenital abnormalities with a wide clinical spectrum, occuring in approximately 1∕5,000–1∕1,500 live births (Endo et al., 1999; Van der Putte, 1986; Tsuda et al., 2011; Levitt & Peña, 2005). Recent studies demonstrated that ARMs are a group of complex malformations affected by many factors, and genetic factors have a great influence on the pathogenesis of ARM, such as Shh, Wnt, BMP-4, Gli2, Hox, Fgf, PCSK5, P63, and others (Wang, Li & Cheng, 2015; Wong et al., 2013; Khanna et al., 2018). Although ARM has been extensively studied, the detailed pathogenesis of ARM is still unclear; in particular, the molecular mechanisms that are involved in the ARMs remain uncharacteristic. It is generally known that non-coding RNAs (ncRNAs), are a family of RNA molecules that regulate gene expression but do not typically code proteins; ncRNAs participate in a variety of cellular processes including cell metabolic function and development (Mattick & Makunin, 2006; Thum & Condorelli, 2015; Godwin et al., 2010; Wang & Chang, 2011). NcRNAs can be subdivided into long ncRNAs (lncRNAs) with a length range from 0.2 and 2 Kb and small ncRNAs (<200 nucleotides long), which encompass microRNAs (miRNAs). In eukaryotic cells, miRNAs can regulate 31% of the target genes and participate in multiple processes of biological development and diseases (Lewis et al., 2013). However, there is a paucity of literature concerning the analysis of miRNA and mRNA co-expression in the development of ARMs. Hence, the purpose of our study was to integrate miRNA-mRNA expression profiles in the terminal hindgut tissue of ARM fetal rats at the twentieth day of the embryo (E20) with using Agilents miRNA and mRNA microarrays. In addition, the relevant data were used for bioinformatics analysis, which includes predicted target genes, GO enrichment analysis, and KEGG pathway analysis.

Materials and Methods

Animals

Ethical approval was obtained from the Capital Institute of Pediatric prior to the study (2015027). In total, 18 healthy and mature SD of similar weight (12 females, six males) were provided by the Beijing Vital River Laboratory Animal Technology Co,. Ltd. The procedures for generating ARMs in fetal rats are described in earlier reports (Bai et al., 2004). The rats were divided into two equal groups randomly. Two female rats were mated to one male rat. The presence of sperm in a vaginal smear was deemed to be gestational day 0 after overnight mating. The pregnant rats were randomly divided into the experiment group and the control group. In the experimental group, six pregnant rats received a single dose of 1% ethylenethiourea (ETU, 125 mg/kg) on gestational day 10. In the control group, the six pregnant rats received a corresponding amount of distilled water on gestational day 10. Embryos were acquired by cesarean delivery on GD 20. The presence of ARMs was identified through the light microscope (see Supplemental Information 1), then the embryos were subdivided into ARM group and normal group. On gestational day 20, 12 pregnant rats of similar weight were acquired by cesarean section, thereby resulting in approximately 100 fetal rats in total. One centimeter of terminal hindgut tissue from the embryos was resected, and frozen in liquid nitrogen immediately to prepare for the extraction of RNA with Trizol methods.

RNA isolation and quantification

Total RNA was isolated with the TRIzol reagent (Invitrogen, Carlsbad, CA, USA) according to the manufacturer’s protocol. Total RNA was quantified by the NanoDrop ND-2000 (Thermo Scientific, Waltham, MA, USA) and the RNA integrity was assessed with using Agilent Bioanalyzer 2100 (Agilent Technologies, Santa Clara, CA, USA, Agilent Bioanalyzer 2100 inclusion criteria: A: RIN ≥7 and 28S/18S ≥0.7, follow-up experiments can be carried out). The sample labeling, microarray hybridization, and washing were performed based on the manufacturer’s standard protocols. Briefly, 5 ug total RNA were dephosphorylated, denaturated and then labeled with Cyanine-3-CTP. After purification the labeled RNAs were hybridized onto the microarray. RNA quantification detail was described in Supplemental Information 2.

Microarray analysis

Three ETU-induced ARM fetal rats and three control rats were chosen for microarray analysis randomly. The Agilent Rat miRNA Microarray (8*15K) and Agilent Rat lncRNA V2 (8*60K) microarray (including mRNA microarray) were utilized to evaluate miRNA and mRNA expression profiles, respectively, in the two groups. Microarray analyses were performed by Shanghai OEBiotech.Co., Ltd, Shang hai, China.

Real-time RT-PCR of miRNA and mRNA

The process of Real-time RT-PCR of miRNA were as follows: each PCR reaction contained 3 µl cDNA, 14 µl mQ-H2O, 3 µl PCR-buffer, 3 µl dNTP solution, 3 µl BSA solution, 1 µl Taq-polymerase, 0.17 µl uracil-DNA glycosylase, and 3 µl of corresponding primer. The reactions were performed under the following conditions: initial denaturation at 94 °C for 2 min, 50 cycles; denaturation at 94 °C for 10 s; and annealing and elongation at 60 °C for 20 s. PCR specificity was controlled by melting curves. In each experiment, one plate contained samples of analyzed cDNA with a primer for the target genes and the reference gene (three repeats each). The relative levels of gene expression were determined as ΔCt = Ct gene −Ct reference, and the fold change in gene expression was calculated with the 2-ΔΔCt method. Experiments were repeated in triplicate.

The process of Real-time RT-PCR of mRNA were as follows: RNA (1 µg) was reverse-transcribed by using the QuantiTect Reverse Transcription Kit (Qiagen, Hilden, Germany) following the manufacturer’s instructions. Each quantitative real-time PCR was performed using 0.3 µl of cDNA in a final volume of 25 µl under the following conditions: 50 °C for 2 min, 95 °C for 10 min, 40 cycles of 95 °C for 15 s, 60 °C for 60 s. A dissociation procedure was performed to generate a melting curve for confirmation of amplification specificity. GAPDH was used as the reference gene. The relative levels of gene expression were determined as ΔCt = Ct gene −Ct reference, and the fold change in gene expression was calculated with the 2-ΔΔCt method. Experiments were repeated in triplicate. The primers sequences for miRNAs and mRNAs can be seen in Supplemental Information 3.

Target prediction and function analysis

Three bioinformatics databases, including Targetscan, microRNAorg and PITA, were utilized to predict the potential target genes. The Target Scan version 7.1 was used to obtain the predicted mRNA targets for each differential expression of miRNA. Then these predicted genes were compared with the mRNA microarray datas, and overlaps of them were determined. The Gene Ontology and KEGG Pathways were considered to elucidate the biological functions of these differentially expressed miRNA target genes, and the cytoscape was adopted to conduct the network analysis.

Statistical analysis

The SPSS version 21.0 package (Chicago, IL, USA) was used for statistical analysis. The independent sample student t test, analysis of variance, and nonparametric test were rationally utilized for the between-group differences. The P value <0.05 was indicated to be significant difference.

Results

miRNA and mRNA differential expression profiles

The miRNA expression profiles were significantly different in terminal hindgut tissues between the ARM and control groups. The process of the microarray and data analysis was shown as follows (Fig. 1). Of all the miRNA measured, 33 miRNAs were identified. Among the differently expressed miRNAs, eight miRNAs were up-regulated in the terminal hindgut tissue of the ARM group, while 25 miRNAs were down-regulated (>2-fold changes, P < 0.05; Table 1). The cluster analysis was shown in Fig. 2. In order to investigate the potential targets of the altered miRNAs, we also determined the mRNA expression profiles of the terminal hindgut tissues by using the Agilent microarray. Greater than 20,000 rat genes and transcripts were investigated and the ARM group and control group were compared. Among the 772 differentially expressed mRNAs, 350 of the genes were significantly up-regulated and 422 were down-regulated (≥2.0 fold changes, P < 0.05) in the ARM group compared with control group. The top 20 upregulated genes and the top 20 downregulated genes were listed in Table 2.

Figure 1 The flow diagram that describe the process of the microarray and data analysis.

Table 1 List of differentially expressed miRNAs between ARM and normal rats.

Upregulated miRNAs	P value	FC (abs)	Downregulated miRNAs	P value	FC (abs)	
rno-miR-221-3p	0.0044	2.836	rno-miR-133a-5p	1.51E–05	128.189	
rno-miR-10b-5p	0.0118	2.653	rno-miR-30c-2-3p	6.21E–04	45.897	
rno-miR-326-3p	0.0129	40.909	rno-miR-144-5p	0.0024	43.322	
rno-miR-183-5p	0.0201	2.305	rno-miR-543-3p	0.0034	67.572	
rno-miR-3084a-3p	0.0259	13.513	rno-miR-136-5p	0.0037	40.949	
rno-miR-200a-3p	0.0266	2.153	rno-miR-136-3p	0.0041	46.235	
rno-miR-3559-5p	0.0352	2.116	rno-miR-376b-3p	0.0042	2.160	
rno-miR-598-3p	0.0495	7.129	rno-miR-486	0.0053	3.118	
			rno-miR-542-5p	0.0054	2.070	
			rno-miR-1-3p	0.0093	10.819	
			rno-miR-133a-3p	0.0143	3.407	
			rno-miR-133b-3p	0.0146	6.663	
			rno-miR-451-5p	0.0172	2.869	
			rno-miR-347	0.0183	2.557	
			rno-miR-1b	0.0192	2.778	
			rno-miR-381-3p	0.0194	2.415	
			rno-miR-503-5p	0.0199	2.719	
			rno-miR-99a-5p	0.0209	2.147	
			rno-miR-495	0.0248	2.758	
			rno-miR-206-3p	0.0300	17.344	
			rno-miR-411-5p	0.0353	2.031	
			rno-miR-128-3p	0.0413	2.567	
			rno-miR-431	0.0424	2.378	
			rno-miR-122-5p	0.0475	212.683	
			rno-miR-99b-3p	0.0490	12.287	

Figure 2 Unsupervised hierarchical clustering of differentially expressed miRNAs in ARM group.

Red indicates higher expression and green indicates lower expression in hindgut tissue of ARM fetal rat. White means no expression difference.

GO and KEGG pathway analysis of the predicted miRNA targets

The Gene Ontology (GO) system was used for functional enrichment analysis of the differentially expressed miRNAs predicted target genes. GO functional annotation of the predicted target genes of the differentially expressed miRNAs was shown in Fig. 3A. According to molecular function analysis, there were 166 GO terms significantly enriched (P < 0.05), and they were predominantly related to protein binding, identical protein binding, etc. in ARM development process. Analysis of the biological processes revealed the 622 significantly enriched GO terms (P < 0.05) to be predominantly associated with transcription, DNA-templated in ARM development process. Cellular component analysis identified 143 significantly enriched GO terms (P < 0.05) to be primarily related to cytoplasm, extracellular exosome in ARM.

Table 2 Top 20 up- and downregulated mRNAs between ARM and normal fetal Rats.

Downregulated mRNAs	Fold change	P value	Upregulated mRNAs	Fold change	P value	
Ky	87.915	0.0178149	Rpap1	40.312	0.0187	
Fitm1	86.903	0.0158851	Alg2	21.131	6.17E–05	
Col9a2	69.592	0.0101831	P2rx4	19.245	0.0133	
Cited1	66.896	0.0132214	Duoxa2	14.719	0.0034	
Cryab	59.710	0.0125412	Cyp4v3	13.758	0.0404	
Serpinf1	52.046	0.0191088	RT1-A1	11.290	0.0312	
Ablim3	50.569	0.0230989	Dhx32	11.101	9.78E-05	
Gatm	46.246	0.0237851	Fgf9	9.546	0.0044	
LOC685612	44.668	0.0173458	Gucy1b3	8.766	3.08E-04	
Col7a1	43.956	0.0096485	Snrnp35	8.671	0.0021	
Sgcg	41.283	0.0067426	RGD1562127	8.643	0.0016	
Ttn	39.483	0.0055567	Kcnk2	8.621	0.0116	
Myoz1	37.311	0.0189420	Aldh1b1	8.390	0.0175	
Gfra4	33.696	0.0098202	Sntg1	8.082	8.35E-04	
Csrp3	32.095	0.0056248	Mogat2	7.291	5.74E-04	
Npepo	30.003	0.0205087	Slpi	7.255	0.0237	
Clec4f	28.918	0.0107149	Adamdec1	6.929	3.22E-04	
Epha3	28.519	0.0070704	Itgal	6.750	0.0342	
Mybpc1	28.450	0.0113402	Madcam1	6.447	0.0352	
Zbtb16	26.716	0.0063771	Tmem151a	6.351	0.0028	

Figure 3 GO and KEGG pathway analysis of differentially expressed miRNAs predicted target genes.

(A) GO functional annotation of differentially expressed miRNAs predicted target genes (top 10), including biological process, cellular component and molecular function. (B) Predicted target mRNAs of differentially expressed miRNAs enriched in the KEGG pathway scatter plot showing the statistics of pathway enrichment in the ARM group.

The potential functional pathways of differentially expressed miRNA targets were then predicted by KEGG pathway system. The DAVID, and KEEG pathway system was used to analysis the microarray data. It showed that 64 pathways were predicted to be significantly related to ARM (P < 0.05). HIF-1 signaling pathway, Cytokine-cytokine receptor interaction, Pathways in cancer, GABAergic synapse, and cAMP signaling pathway were the main pathways involved in these predicted pathways (Fig. 3B).

GO and KEGG pathway analysis of the differentially expressed mRNAs

The Gene Ontology (GO) system was used for functional enrichment analysis of the differentially expressed mRNAs. The top 10 GO pathway enrichment terms for differentially expressed mRNAs were shown in Fig. 4A. According to molecular function analysis, there were 68 GO terms significantly enriched (P < 0.05), and they were closely related to heparin binding, actin binding, tropomyosin binding, etc. in ARM development process. According to biological processes analysis, it revealed that 331 significantly enriched GO terms (P < 0.05) to be predominantly associated with muscle contraction, muscle organ development, cardiac muscle contraction, and regulation of muscle contraction in ARM development process. Cellular component analysis identified 56 significantly enriched GO terms (P < 0.05) to be primarily related to extracellular space, myofibril, and troponin complex in ARM.

Figure 4 GO enrichment and KEGG pathway analysis of differentially expressed mRNAs.

(A) Top 10 pathway enrichment terms for differentially expressed intersection mRNAs, including biological process, cellular component and molecular function. (B) KEGG pathway analysis of the differentially expressed mRNAs was performed. Top 30 pathway enrichment terms for differentially expressed intersection mRNAs.

The KEGG pathways for the differentially expressed mRNAs were then performed by KEGG pathway system. The DAVID and KEEG pathway system were used to analysis microarray data. It revealed that 53 pathways were predicted to be significantly related to ARM (P < 0.05). Dilated cardiomyopathy, Hypertrophic cardiomyopathy (HCM), Cell adhesion molecules (CAMs), and ECM-receptor interaction were the main pathways involved in the development process of ARM (Fig. 4B).

miRNA target gene prediction

As miRNAs negatively regulate mRNAs expression via degradation or translation inhibition. We intersect the predicted target mRNAs of upregulated miRNAs with the downregulated mRNAs from microarray, and the predicted target of mRNAs of downregulated miRNAs with the upregulated mRNAs from microarray. As a consequence, 25 miRNAs and 76 genes formed 142 miRNA-target gene pairs with an inverse correlation of expression. Among the 422 significantly decreased mRNAs, 33 mRNAs were the predicted targets of the five upregulated miRNAs. Similarly, 43 in 350 overexpressed mRNAs and 20 downregulated miRNAs were predicted as targets. According to the above method, a total of 142 significant miRNA-mRNA pairs were predicted, consisting of the 25 differentially expressed miRNAs and 76 mRNAs. The target prediction of eight miRNAs isn’t available in miRWalk databases.

Validation of miRNA and mRNA expression profile by RT-qPCR

To further investigate the credibility of our microarray based results, qRT-PCR was performed in 12 pairs of matched ARM/control fetal rat terminal hindguts. From the differentially expressed miRNAs and mRNAs observed by microarray, five miRNAs (down: rno-miR-1-3p, rno-miR-99b-3p, rno-miR-206-3p, Up: rno-miR-598-3p, rno-miR-3084a-3p) and five mRNAs (down: Fgf16, Wnt16, Cdh15, Up: Trim9, Ccl7) were chosen for RT-PCR validations. Relative expression levels of the selected miRNAs and mRNAs were depicted in Figs. 5A & 5B. The qRT-PCR results confirmed the accuracy of microarray findings. Compared with controls, rno-miR-598-3p and rno-miR-3084a-3p were significantly increased, while rno-miR-1-3p, rno-miR-99b-3p, and rno-miR-206-3p were significantly decreased in the terminal hindgut tissues of ARM fetal rat. Besides, qRT-PCR analysis revealed that the expressions of Trim9, Ccl7 were upregulated, whereas Fgf16, Wnt16, and Cdh15 were downregulated in the terminal hindgut tissues of ARM fetal rat, which is in line with the microarray data, too. Although the magnitude of changes differed between the two methods, these results demonstrated a high consistency between the microarray and RT-qPCR.

Figure 5 qRT-PCR validation of differentially expressed miRNAs and mRNAs.

Quantitative reverse transcription was performed to confirm the expression of five selected miRNA (A) and five mRNA (B).

The regulatory network of miRNAs and target genes

The miRNA-target gene pairs via Cytoscape software were used to construct the miRNA-target genes regulatory network. Using the 142 miRNA-target gene pairs, a miRNA-target gene regulatory network was constructed (Fig. 6). In this network, rno-miR-128-3p, rno-miR-200a-3p, rno-miR-30c-2-3p, rno-miR-495, and rno-miR-543-3p, which regulate 15, 12, 13, 15 and 13 targets, respectively, demonstrated the highest connectivities. Whereas Igf1, Fut4, Kcnk2, Ngfr, and Nap1l2, which were regulated by eight, six, four, four and four miRNAs, respectively, were the mRNAs with the highest connectivities. According to the the network analysis, most of the identified transcripts were connected with several miRNAs, it suggested that multiple miRNA-mRNA interactions combinatorial effect in gene regulation by coexpressed endogenous miRNAs in ARMs.

Figure 6 The regulatory network between miRNAs and target genes in ARM.

The round and square represent the mRNAs and miRNAs, respectively. The purple and yellow colors represent the relatively low and high expression, respectively. The large geometric drawing indicates the more miRNAs or genes interacted with it.

Discussion

ARMs have been reported to be a multigene genetic disease. Both genetic and environmental factors are involved in the etiopathogenesis of ARMs. The spectrum of ARM phenotypes range from stenotic anus to cloacal malformation, and the etiology of ARMs are still unknown (Wang, Li & Cheng, 2015). The development of hindgut in the embryonic period is governed by multiple genes in the relevant signaling pathways. A result of genetic change in any stage of the hindgut development may lead to varieties of ARM phenotypes. Recent studies demonstrated that molecular and genetic facors were the contributing roles in its etiology. Only when the cascades of relevant genes expression are well orchestrated can the normal anorectum develop. Hence, further intensive study of ARMs may be benefical for illuminating the etiopathogenesis of this complex congenital malformation.

Studies of microRNAs (miRNAs) indicated that they are a series of noncoding RNAs with a length of 22 nucleotides, have the function of mediating silence and post-transcriptional regulation of gene expression (Ambros, 2004; Bartel, 2004), and are involved in many vital processes, including proliferation, cell differentiation, migration and apotosis (Hosako et al., 2009). The principle of miRNAs is that the miRNA usually interacted with the 3′ untranslated regions (3′-UTRs) of their targets mRNAs to negatively regulate gene expression at the post-transcriptional level (Saito et al., 2009). Up to now, there were more than 2,500 human miRNAs recorded in the miRBase (Kozomara & Griffiths-Jones, 2011; Kozomara & Griffiths-Jones, 2014), and research showed that more than 60% of protein-coding transcripts were predicted to be the targets of miRNAs and regulated by miRNAs (Fabian, Sonenberg & Filipowicz, 2010). However, there is paucity of literature related to the microRNA expression profiles and their biological effects in ARMs.

In the present study, we evaluated the terminal hindgut expression of miRNAs and mRNAs in both ETU-induced ARM rats and control rats using microarray analysis. We found 33 miRNAs and 772 mRNAs differentially expressed in hindgut tissues between the ARM fetal rats and the controls. GO terms and KEGG pathway analysis for the differentially expressed mRNA further revealed that the miRNA-mRNA contributed a lot to the development of ARMs. Seventy-six potential target genes for the 25 miRNAs were predicted by intersecting three bioinformatics databases and differentially expressed mRNAs, resulting in 146 miRNA-mRNA pairs. Morever, the expression of several miRNAs and mRNAs were validated by qRT-PCR and the outcomes were highly in accordance with the microarray data, which confirmed the credibility of our microarray data.

Among the 33 miRNAs that were differently expressed in the hindgut tissue of ARM fetal rats according to the miRNA microarray data, previously studies have reported that many of these miRNAs (rno-miR-221-3p, rno-miR-200a-3p, rno-miR-1-3p, rno-miR-133b-3p, rno-miR-133a-3p, rno-miR-486, rno-miR-451-5p) identified in our result were associated with many disease such as colorectal disease, prostate and pancreatic cancers (Sun et al., 2011; Qin & Luo, 2014; Kneitz et al., 2014; Sarkar et al., 2013). Kneitz et al. (2014) reported that miR-221 is an oncogenic miRNA which targeted CD117, and that it has the function of preventing cell proliferation and migration in endothelial cells. In this study, we also found that miRNA-200a-3p was significantly up-regulated; Pichler et al. (2014) reported that miRNA-200a-3p is able to exert its regulatory effect on EMT (epithelial to mesenchymal transition) and is involved in cancer stem cell properties in colorectal cancer.

miRNAs exert their function of regulating through inhibiting or degrading the translation of its targets. Hence, it is extremely important to identify miRNA target genes to clarify the development process of ARM. It is well known that the primary cause of ARMs was mainly attributed to the gene expression abnormalities. Gene mutations have been reported to be the most important contributing roles in the pathogenesis of ARMs. The normal development of hindgut will be disturbed by the mutations of the related genes, which thus lead to a series of ARM phenotypes.

Among the 772 mRNAs that showed to be significantly dys-regulated according to microarray datas, many of these mRNAs (Wnt10a, Wnt16, Fgf7, Fgf9, Fgf16, Fgf2, Bmp3, Hoxd1) identified in our result have been reported previously to be associated with the pathogenesis of ARMs. Based on GO analysis of biological processes and pathways, it revealed that the 772 significantly dys-regulated genes to be mainly relate to muscle contraction, muscle organ development, regulation of muscle contraction, skeletal muscle tissue development, response to drug, positive regulation of myotube differentiation, and these cellular events are closely related to extracellular space, myofibril, extracellular matrix, and contractile fiber in ARM development process. KEGG pathway enrichment analysis revealed that these differentially expressed mRNAs were potentially associated with the cell adhesion molecules (CAMs), ECM-receptor interaction, focal adhesion, PI3K-Akt signaling pathway, and etc. The PI3K-Akt signaling pathway is activated by many types of cellular stimuli or toxic insults and regulates fundamental cellular functions such as transcription, translation, proliferation, growth, and survival. The binding of growth factors to their receptor tryosine kinase (RTK) or G protein-couple receptors (GPCR) stimulates class Ia and Ib PI3K isoforms, respectively. PI3K-Akt signaling pathway containing many genes including Colla1, Fgf7, Fgf2, Fgf16, Fgf9, Igf7, Prkzc, etc., while some of those genes are also implicated in the development process of ARMs. These genes linked to the key pathways such as Shh, Fgf, and Wnt/β-catenin signaling pathways or were involved in the regulation of their downstream targets such as Bmp4, Fgf10, Gli2, Gli3, and Wnt5a to trigger the abnormal development of anorectum of ARMs.

In order to further investigate the role of miRNAs in ARMs, we integrated miRNA and mRNA expression profiles with miRNA target predictions by the miRWalk database to identify the positively and negatively correlated miRNA/mRNA pairs. Consequently, 25 miRNAs and 76 genes formed 146 miRNA-target gene pairs displaying reciprocal level of expression. The target prediction of eight miRNAs is not available in miRWalk databases. The negative correlation between miRNAs and their predicted target mRNAs expression identified in our study supports the hypothesis that miRNAs significantly regulate gene expression in the pathogenesis of ARMs. Although some miRNAs and their taget mRNAs did not display negative correlation, their regulatory effect might have been concealed by additional regulatory mechanisms or remain unclear. Hence, only the miRNAs and their predicted target mRNAs that displayed significant correlation with their cognate miRNAs expression were analysed acccording to our algorithm. In the present study, through detecting the predicted target genes of the differentially expressed miRNAs, we found that some important genes were involved in the pathogenesis of ARM, i.e., Fgf, Hox, Bmp, Wnt, and others. Recently, a study (Jin et al., 2017) showed that the miR-193 was significantly upregulated in the terminal hindgut tissues of ARM fetal rat, and miR-193 inhibited the expression of Hoxd 13 in ARM fetal rat by targeting Hox, while the normal expresion of Hoxd is absolutely essential to the development of the anorectum.

Fgf signaling is important during the early embryonic development in vertebrates (Böttcher & Niehrs, 2005; Gambarini et al., 1996); it plays a key role in patterning the gut tube through promoting posterior and inhibiting anterior endoderm cell fate (Kondoh, Kobayashi & Nishida, 2003). Fgf is expressed in epithelial cell, and has the function of patterning the gut tube by promoting posterior and inhibiting anterior endoderm cell fate (Gambarini et al., 1996). Fgf16 encoded a member of a family of proteins which possessed broad mitogenic and cell survival activities, and were associated with a variety of biological processes such as embryonic development, tumor growth and invasion, cell growth, morphogenesis, tissue repair. In addition, as predicted by bioinformatics databases, Fgf was the potential target gene of rno-miR-221-5p, rno-miR-133b-3p, rno-miR-381-3p, rno-miR-431, and rno-miR-495. The Wif1 (Wnt inhibitory factor 1) gene, which mediates the activity of Wnt (Wingless drosophila integration sites) signaling pathway, is a predicted target gene of the rno-miR-451-5p and rno-miR-495 according to TargetScan 7.1. Wif1 is an inhibitory Wnt (Wingless drosophila integration sites) that negatively regulates Wnt signaling. This gene is a member of the Wnt gene family, it plays a key role in the regulation of cell fate and patterning during embryogenesis (Jönsson & Andersson, 2001). The Wnt was reported to be a crucial signaling pathway and it has a paramount regulatory effect during the development of anorectum. Bambi (BMP and activin membrane-bound inhibitor) gene was significantly upregulated in terminal hindgut tissue of ARM fetal rats in our results, and is a predicted target gene of rno-miR-128-3p which was significantly downregulated. This gene encodes a transmembrane glycoprotein related to the type I receptors of the transforming growth factor-beta (TGF-β) family, whose members play important roles in signal transduction in many pathological processes, the encoded protein may function to limit the signaling range of the TGF-β family during early embryogenesis. Our results showed that there existed an obviously negative correlation between the expression of rno-miR-128-3p and Bambi during the development of ARMs.

The expression of rno-miR-221-3p, rno-miR-381-3p, rno-miR-128-3p, rno-miR-495, and rno-miR-221-3p were altered in the terminal hindgut in ARM fetal rats according to our results, and those miRNAs all involved in feed-forward loops that amplify or inhibit the Fgf, Bmp, Wnt signaling in the pathogenesis of ARMs. In the present study, a series of genes such as Fgf, Wnt that is necessary for embryonic development were low expressed, and they found to be related to the abnormal expression of rno-miR-221-3p, rno-miR-381-3p, rno-miR-128-3p, rno-miR-495, and rno-miR-221-3p in terminal hindgut tissue of ARM fetal rats. Therefore, we proposed that, in the process of ARMs, the dys-regulated expression of Fgf, Wnt mediated by relevant miRNAs, such as rno-miR-221-3p, rno-miR-381-3p, etc. would disturb the cytokine-cytokine receptor interaction, cell growth, cell differential during embryogenesis and promote abnormal development of anorectum.

It should be noted that because of the temporal and spatial expression pattern of mRNAs during the hindgut development in rat embryos with ETU-induced ARMs, there are some differences in our study in comparison with other studies. Also, some limitations existed in our study. On one hand, the expression of mRNAs are influenced by multiple factors. Both the upstream miRNA and the neighbour genes can affect the gene expression, and this condition is not taken into consideration in our work and can be integrated in future research. Also, the miRNA/mRNA that were predicted to have a potential effect on influencing the expression of pathogenic genes need further functional experiments for validation.

Conclusion

In this work, we identified the differentially expressed miRNAs and mRNAs in the terminal hindgut between ARM fetal rats and normal ones. We strongly suggest that miRNAs are paramount in regulating gene expression in the pathogenesis of ARM. The present work can be regarded as a new perspective and direction for future research on ARMs.

Supplemental Information

Table S1 Case VS control cluster data

Click here for additional data file.

Table S2 Case VS control 2 databases

Click here for additional data file.

Table S3 Case VS control 2 databases—biological process

Click here for additional data file.

Table S4 Case VS control 2 databases—cellular component

Click here for additional data file.

Table S5 Case VS control 2 databases—molecular function

Click here for additional data file.

Table S6 Case VS control 2 databases—KEGG

Click here for additional data file.

Table S7 Case VS control differential screening

Click here for additional data file.

Supplemental Information 1 The appearance of the anus of the ARM group and control group

Click here for additional data file.

Supplemental Information 2 Sample quality inspection report

Click here for additional data file.

Supplemental Information 3 Primers sequences for miRNAs and mRNAs

U6 and GAPDH were used as the internal controls for miRNAs and mRNA, respectively.

Click here for additional data file.

Additional Information and Declarations

Competing Interests

Author Contributions

Animal Ethics

Microarray Data Deposition

Data Availability

The authors declare there are no competing interests.

Hui Xiao conceived and designed the experiments, performed the experiments, analyzed the data, contributed reagents/materials/analysis tools, prepared figures and/or tables, authored or reviewed drafts of the paper, approved the final draft.

Rui Huang performed the experiments, analyzed the data, contributed reagents/materials/analysis tools, prepared figures and/or tables, authored or reviewed drafts of the paper, approved the final draft.

Mei Diao performed the experiments, analyzed the data, contributed reagents/materials/analysis tools, authored or reviewed drafts of the paper, approved the final draft.

Long Li conceived and designed the experiments, performed the experiments, authored or reviewed drafts of the paper, approved the final draft.

XiaoDai Cui conceived and designed the experiments, performed the experiments, contributed reagents/materials/analysis tools, approved the final draft.

The following information was supplied relating to ethical approvals (i.e., approving body and any reference numbers):

Ethical approval was obtained from the Capital Institute of Paediatric prior to the study (2015027).

The following information was supplied regarding the deposition of microarray data: GEO 120445: https://www.ncbi.nlm.nih.gov/geo/query/acc.cgi?acc=GSE120445.

The following information was supplied regarding data availability:

The raw data are provided in the Supplemental Files.

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
