# Peer review of "Integrative analysis of microRNA and mRNA expression profiles in fetal rat model with anorectal malformation"

_PeerJ, doi:10.7717/peerj.5774_

## Round 0.1 · original submission · Major Revisions

Please review the four reviews for your manuscript, they have many good suggestions for improvements.

Reviewer 1 ·

Basic reporting

No Comments

Experimental design

No Comments

Validity of the findings

No Comments

Additional comments

In this study, entitled "Integrative analysis of microRNA and mRNA expression profiles in the fetal rat model with anorectal malformation" authors interrogate the miRNA and mRNA expression profiles in terminal hindgut tissue of 20d old ARM rat embryos. Based on the data provided, authors suggest that they identified the differentially expressed miRNA and mRNA in terminal hindguts of ARM rat samples compared with control and list the various pathways and other genomic features involved in such samples. The study is significant in determining the miRNA expression in ARM samples, however following things need to be addressed:

1: Authors should be descriptive about qPCR in method section – like How much cDNA was used? What was the PCR conditions? How many amplification cycles used? Whether melt curve analysis was performed? If yes, mention the same in the method section. Also, provide a relevant reference for qPCR analysis.

2: Authors should provide PCA plot for every 3 replicates of case and control microarray. It would be really helpful to distinguish the case and control replicates.

3: Authors have mentioned that they identified ARM by light microscopy. Include those images as a part of Figure 1.

4: Do authors determined the sex of embryos used for study? Mention it in the method section.

5: Why authors have chosen 20d gestation age? No citation provided. Must reference it.

6: Also, authors should provide the reference for ETU usage for inducing ARM. Presently, it is missing.

7: Provide the list of primer sequence used for qPCR analysis as a supplementary table.

8: Authors should be consistent in writing miRNA throughout the manuscript. At many places it is MiRNA. Same should be corrected.

9: There are grammar and spelling errors in the manuscript that should be addressed.

Reviewer 2 ·

Basic reporting

The author described an integrative analysis of microRNA and mRNA profiles in rat model of anorectal malformation by miRNA array, 33 miRNAs and 772 mRNAs were then screened between ARM group and control group,providing a basis for further research on miRNAs and mRNAs in the pathogenesis of ARMs.

Experimental design

There was a severe flaw in the experimental design. All the fetal samples used in this study were obtained on the gestional gay 20, on this time point, development of the hindgut has finished as we know that the gestional day 13-14 was the key time point for anorectal development, microRNA and mRNAs can only play their roles during this period, after that, parts of microRNA and mRNAs may lose their activity. therefore, author should choose the fetal samples on the gestional day 13-14 but not 20.

Validity of the findings

the results were valid but not scientific.

Additional comments

The author described an integrative analysis of microRNA and mRNA profiles in rat model of anorectal malformation by miRNA array, 33 miRNAs and 772 mRNAs were then screened between ARM group and control group,providing a basis for further research on miRNAs and mRNAs in the pathogenesis of ARMs. Although the results were attractive, there was a severe flaw in the experimental design leading the results were not scientific. there was so many spelling and grammatic errors throughout the paper making it difficult to read

Reviewer 3 ·

Basic reporting

Needs work. I believe that the article uses clear English to describe the study design and setup but could use some work on the following aspects
1) There are a lot of spelling and grammatical errors that can be resolved with a spell check and some minor proofreading. (Some of these are marked in my copy of the manuscript, but I have not marked it up thoroughly)
2) Additionally, the authors switch between present and past tense frequently especially in the discussion making it harder to understand distinguish statements of fact from speculation.
I believe that the authors have not given sufficient background of the ETU induced ARM model. Is this a well-established model in the field? Are there other models available to study ARM? If so, the authors should cite relevant studies and describe their reasons (and pros/cons) of selecting this particular model. Additional details including dose of ETU used and days of gestation before analysis will help flesh out the model in more complete details.
The authors have done a good job of sharing all the raw data. Are the planning to upload the microarray data for public access on NCBI GEO or other suitable servers?
Additionally, I believe that figure 1 can be replaced by a more informative study design. The number of animals used for analysis for mRNA vs miRNA analysis for example remains unclear to me. This would help in clearly conveying the work rather than a workflow which is fairly standard and easily grasped by perusing the materials and methods.

Experimental design

I believe that the current work is original primary research that is within the aims and scope of the journal.
I commend the authors for clearly defining the research question at hand and explaining its relevance to the understanding of ARM
I believe that the authors have attempted to clearly explain the methodology used although i am confused by the number of animals used per arm of the study. The animal section of materials and methods states that 20 mice were used for the ETU arm of the study and an unknown number were included in the control group. Additionally of the 20 mice, 12 were used for total RNA and 6 for miRNA. What were the 2 unreported mice used for? Also, how were the numbers (12 and 6 chosen for the mRNA and miRNA analysis respectively? Did you run power calculations? If not, what was the basis for this division?) My recommendation to clarify this would be to create a schematic depicting the study design along with animal numbers

I believe that the authors have thoroughly investigated the mRNA and miRNA profiles obtained by the microarrays and subsequent analysis. However, the genes they chose for qPCR validation are not the top changed genes on their list. For example, mir-598 is the lowest gene on the up regulated miRNA list while mir-3084a is somewhere in the middle. What is the rationale behind this specific choice of miRNA for validation? Why not just pick the top 5 per category? The choice of mRNA for validation is similarly puzzling.
I would also like for the authors to provide the primer sequences that they used to investigate each gene by qPCR. This will help others who investigate the same genes to contextualize the author’s findings.

Validity of the findings

I believe that the findings in this study are valid. The data used for this study is robust although i believe that additional validation either in the form of protein expression or in vitro analysis on cell lines would certainly bolster the findings. The authors have focused on the RNA expression levels and have done preliminary validation of a handful of genes by qPCR. However, mRNA expression levels without the corresponding impact on the protein levels does not lend itself to drawing meaningful conclusions about the biological question under study.
Additionally, the authors spend a majority of the discussion speculating about the modulated miRNA and their potential targets and attempting to connect it to known biology without much data to support this speculation. While some speculation is clearly justified to create a complete picture, their use of it is excessive and tends towards overstating their findings.

Additional comments

I believe that the authors have made a strong attempt to understand the RNA expression underlying ARM development in the ETU model.
While the authors have characterized the changed genes in the ARM model extensively using KEGG and GO analysis, I believe that more meaning could be obtained by validating some of the changes at a protein level as well.
Additionally, while correlating the miRNA changes with their targets, the authors often use qualitative terms instead of quantitative. Some options to improve this would be by running correlations on the miRNA and their targets to quantify the relationship. Similarly, the authors state that “there is a high consistency between the microarray and qPCR data” (lines 216-217). This can also be quantified.
Overall, I think that the readability and applicability of the paper will be greatly improved by focusing the discussion on the key findings of the paper and its implication. There are several instances where the authors have gone into an extensive discussion of the biology of signaling in ARM although they have personally not investigated any signaling related aspects at all

Annotated reviews are not available for download in order to protect the identity of reviewers who chose to remain anonymous.

Reviewer 4 ·

Basic reporting

In this manuscript the authors evaluated the microRNA and mRNA profile of anorectal malformations (ARM) in fetal rats. The authors for the first time evaluated the microRNA profile in ARM and adds valuable information to the field.

The language should be once proof read for vocabulary and grammatical mistakes.

The discussion part appears to be redundant and should be shortened to make it clear and to the point.

Experimental design

1. Number of Rats given ETU treatment is confusing. As total 20 females were selected for experiment. How many are given ETU treatment and how many are control rats should be made clear.
2. For validation experiments, why particular miRNA and mRNA were chosen should be described? Why highly upregulated and downregulated miRNA and mRNA were not checked with qRT-PCR. miRNA and mRNA showing high connectivity should have been chosen for validation.

Validity of the findings

No comments

Additional comments

The following should be considered to improve the manuscript.
Language should be checked
Line 54: Day 20
Line 55: Spelling “terminal”
Line 76: group “of”
Line 79: correct “pathology”
Line 89: “to the best of my knowledge” should be removed
Line 93: correct “data”
Line 113: RNA isolation and quantification should be described in detail. Homogenization step, RIN number details etc should be provided.
Line 129: internal control for miRNA should have been smaller RNA. Any justification
Line 149: what was the total number of miRNA scanned, upregulated and downregulated should also include % of total in form of vein diagram to make it more clear and representable.
Line 196: there is no English word “Resultly”
Line 330: target prediction cannot be done based on downregulation of rno-miR-381-3p and Fgf16 as they are downregulated among other differentially expressed genes and miRNA. Please change the sentence. This can be done experimentally on one to one basis or predicted bio-informatically.
Fig 1 “Dephosphorylation”
Fig 5: How normalization was done in both cases. Are they comparable??

---

## Round 0.2 · accepted · Accept

Thank you for addressing the reviewers comments.

# Reviewer 1 ·

Basic reporting

No comments

Experimental design

No comments

Validity of the findings

No comments

Additional comments

Authors have done a good job in addressing this reviewer concerns. No more questions.

Reviewer 3 ·

Basic reporting

I believe the authors have taken effort to correct a lot of the errors previously highlighted in this section. A few errors remain but can be fixed by a thorough proofreading of the the final manuscript. a few examples (but not all) of errors listed below
1) recent not rencent on line 72
2) figure 2 legend: black and NOT white means no difference

Experimental design

I believe that the authors have taken effort to correct all the errors previously highlighted in this section. No further comments

Validity of the findings

I believe that the findings in this study are valid.

Additional comments

I have one main request of the authors: Please rework your discussion to only the pertinent points and shorten it dramatically. Currently about half the discussion is redundant to the results section or is irrelevant. The authors have spent several lines purely restating the results from the results section and increased the length of the discussion quite a bit (see lines 288-295, 299-303,319-341 etc). Additionally, the authors have gone into an extensive discussion of the biology of signaling in ARM and into details of micro RNA mediated RNA regulation of several transcripts which i find unnecessary. I believe that this section needs real effort to focus it solely on the findings and implications of the authors findings to prevent it from being a review of the signaling and biology related to ARM.

Reviewer 4 ·

Basic reporting

Authors have included more information and made the manuscript crisp and clear.

Experimental design

ok

Validity of the findings

no comment

Additional comments

We thank the authors for improving and providing additional data to the manuscript for easy understanding.